# The Wild Carrot (*Daucus carota*): A Phytochemical and Pharmacological Review

**DOI:** 10.3390/plants13010093

**Published:** 2023-12-27

**Authors:** Jana Ismail, Wassim N. Shebaby, Joey Daher, Joelle C. Boulos, Robin Taleb, Costantine F. Daher, Mohamad Mroueh

**Affiliations:** 1Pharmaceutical Sciences Department, School of Pharmacy, Lebanese American University, Byblos 1102-2801, Lebanon; jana.ismail@lau.edu (J.I.); mmroueh@lau.edu.lb (M.M.); 2Gilbert and Rose-Marie Chagoury School of Medicine, Lebanese American University, Byblos 1102-2801, Lebanon; joey.daher@lau.edu; 3Institute of Pharmacy and Biomedical Sciences, Department of Pharmaceutical Biology, Johannes Gutenberg University, 55128 Mainz, Germany; joboulos@uni-mainz.de; 4Department of Natural Sciences, School of Arts and Sciences, Lebanese American University, Byblos 1102-2801, Lebanon; robin.taleb@lau.edu.lb (R.T.); cdaher@lau.edu.lb (C.F.D.); 5Alice Ramez Chagoury School of Nursing, Lebanese American University, Byblos 1102-2801, Lebanon

**Keywords:** *Daucus carota*, wild carrot, terpenes, anticancer, anti-inflammatory, antimicrobial

## Abstract

*Daucus carota* L., a member of the Apiaceae family, comprises 13 subspecies, with one being cultivated (*D. carota* L. ssp. sativus (Hoffm.) Arcang.) and the remaining being wild. Traditionally, the wild carrot has been recognized for its antilithic, diuretic, carminative, antiseptic, and anti-inflammatory properties and has been employed in the treatment of urinary calculus, cystitis, gout, prostatitis, and cancer. While extensive literature is available on the phytochemical, pharmacological, and therapeutic evaluations of the cultivated carrot, limited information has been published on the wild carrot. A thorough search was conducted on the phytochemical composition, folk-medicine uses, and pharmacological properties of wild carrot subspecies (*Daucus carota* L. ssp. carota). Various electronic databases were consulted, and the literature spanning from 1927 to early 2023 was reviewed. Thirteen wild *Daucus carota* subspecies were analyzed, revealing over 310 compounds, including terpenoids, phenylpropenoids, flavonoids, and phenolic acids, with 40 constituting more than 3% of the composition. This review also highlights the antioxidant, anticancer, antipyretic, analgesic, antibacterial, antifungal, hypolipidemic, and hepato- and gastroprotective properties of wild carrot subspecies. Existing in vitro and in vivo studies support their traditional uses in treating infections, inflammation, and cancer. However, further research on other subspecies is required to confirm additional applications. Well-designed preclinical and clinical trials are still necessary to establish the safety and efficacy of wild *Daucus carota* for human use.

## 1. Introduction

*Daucus carota* belongs to the family *Apiaceae* or *Umbelliferae* and is part of the *Daucus* genus, which is a polymorphic taxon of multiple species and subspecies [1]. This family is characterized by aromatic, flowering, and celery/parsley-like plants that include edible as well as toxic species. *Daucus carota* L., commonly known as carrot, is classified into twelve subspecies; *D. carota* ssp. *sativus* (Hoffm.) Arcang. (the most well-known variety) and *D. carota* ssp. *boissieri* (Schweinf.) H. A. Hosni (red carrot) are the cultivated and edible varieties [2]. The wild subtypes of the carrot include the subspecies *carota*, *maritimus*, *major*, *maximus*, *gummiferi*, *hispanicus*, *commutatus*, *fontanesii*, and *bocconei* [3]. Selective breeding of an ancestral wild form of carrots, *D. carota* ssp. *carota*, over the centuries led to the origination of a cultivated form of *D. carota*, later made to be known as *D. carota* ssp. *sativus* [4]. Most of the available literature focuses on this specific subspecies, and it has been a topic of interest for researchers since the day of its cultivation. Differentiation among *D. carota* L. subspecies is difficult as the carrot is an outcrossing species, and both cultivated and wild carrot samples coexist, thus presenting the possibility of hybrids’ formation. Nonetheless, Thellung classifies the subspecies into two groups, *eucarota* and *gummiferi*, each classified into five more subspecies. The *eucarota* group includes the subspecies *sativus*, *carota*, *maritimus*, *major*, and *maximus*; these plants are mostly annual or biennial. Plants of the *gummiferi* group, on the other hand, are perennials and include the subspecies *commutatus*, *hispanicus*, *fontanesii*, *bocconei*, and *gummifer* [3], among others.

Multiple studies have been conducted to investigate the chemical composition of the different *D. carota* subspecies, which revealed the abundance of terpenes, phenolics, and flavonoids [5,6,7,8]. The chemical composition of the plant extracts differed between multiple subspecies as well as among plants of the same subspecies. According to Maxia et al. [6], the geographical origin of the plant sample, the plant organs, as well as its stage of development may affect its chemical composition. However, it was generally noted that monoterpenes and/or sesquiterpenes are mostly present in leaves, stems, and blooming umbels [6,7,9].

Traditionally, the wild carrot was utilized by the ancient Greeks and Romans for various medicinal purposes. Dioscorides, a Greek physician and botanist who lived in the 1st century AD, documented the uses of the plant in his renowned work “De Materia Medica” [10]. The plant is known to possess antilithic, diuretic, carminative, antipyretic, analgesic, anti-inflammatory, antiseptic, and anti-diabetic properties, among others [5,11,12,13]. Over the past two decades, the potential medicinal benefits of *D. carota* became clear, as its antioxidant, anticancer, anti-inflammatory, gastroprotective, hepatoprotective, antibacterial, and antifungal activities have been confirmed in-vitro and in-vivo as well [2,14].

Many review articles have been published about cultivated edible carrots, but none has been conducted on wild carrot subspecies. Therefore, the present article aims to present a comprehensive overview concerning the traditional uses, the phytochemical composition, and the pharmacological properties of known wild *D. carota* subspecies. In addition, the potential therapeutic uses of the wild carrot will be evaluated on the basis of the available pharmacological findings.

## 2. Methodology

A thorough search was conducted on the phytochemical composition, folk medicine applications, and pharmacological properties of wild carrot (*Daucus carota*) subspecies. Various electronic databases were consulted such as, PubMed, MDPI, Google Scholar, Science Direct, Elsevier, Springer, Taylor and Francis, Scopus, ACS publications, Wiley On-line Library, and the Lebanese American University Libraries’ online database. These databases collectively offer a comprehensive and extensive range of articles available in the literature on this plant. To perform the bibliographic research, specific keywords were employed such as the wild carrot, *D. carota* ssp. *Carota*, *D. carota* ssp. *Hispanicus*, *D. carota* ssp. *hispidus*, *D. carota* ssp. *gummifer*, *D. carota* ssp. *hispanicus*, *D. carota* ssp. *maximus*, *D. carota* ssp. *maritimus*, and *D. carota* ssp. *major.* The emphasis of this review encompasses phytochemical composition and pharmacological properties including antioxidant, anticancer, anti-inflammatory, antibacterial, and antifungal activities as well as other ethnopharmacological practices. The data pertinent to this study were collected through a comprehensive review of 100 scholarly articles, 25 books, 2 encyclopedias, 2 conference papers, 2 web pages, and 1 meeting report, spanning the years 1927 to 2023. The inclusion criteria in this study were based on the credibility of the sources from which the data were collected. Only data obtained from sources with a high level of credibility were included, emphasizing the importance of dependable and authoritative information to support the study’s findings. The gathered information corresponds to research studies conducted across various global locations, spanning the United States, Europe, the Mediterranean region, North Africa, the Middle East, Western Asia, and China.

## 3. Plant Description and Distribution

The wild carrot, commonly known as Bird’s Nest, Bishop’s Lace, and Queen Anne’s lace [15], is generally a biennial plant (around 1 m height) with distinctive clusters of tiny white flowers that bloom during warm seasons. Unlike the cultivated carrot, the wild carrot has a thin tap root, hairy stems, two- to three-pinnate leaves, pinnate bracts below a concave umbel, when fruiting, and spiny fruits [1]. Its flowering period spans from May to September [16].

It is important to mention that due to the similarities in their appearances, incidences of confusion have been observed between *D. carota* and poison Hemlock (*Conium maculatum*), which is infamously linked to the death of the Greek philosopher Socrates [17,18]. Thus, providers and consumers should be able to identify and distinguish between the two plants. For example, *D. carota* stems and leaves are hairy, whereas *Conium maculatum* stems are smooth and display purple blotching. Regarding the aerial parts, *D. carota* has a purple/red flower at the center of the umbel, absent in *Conium maculatum*, and the umbels in *D. carota* appear flatter, resembling an umbrella in *Conium maculatum*. Finally, only *D. carota* demonstrates three-pronged bracts at the base of the flower/umbel. Despite their similar fern-like leaves, extreme caution should be exercised when identifying them [19].

*D. carota* L., which is originally native to Europe and southwestern Asia, grows in a temperate environment, as it can be usually spotted in Europe (Germany, England, Austria, Poland, Greece, Sweden, and Hungary), Africa (Tunisia, Egypt, and Mauritius), America (Canada, Puerto Rico, and the USA), and Asia (Iraq, Iran, Jordan, and Lebanon) [20].

## 4. Ethnopharmacological Use of *Daucus carota*

About 300,000 species of higher plants have been identified worldwide, of which 17,810 species have been documented by The Royal Botanic Gardens, Kew, as medicinal plants (State of the World’s Plants, 2016). According to the World Health Organization (WHO), eighty percent of Africans utilize different forms of herbal medicine [21,22]. The global annual market of these herbal products is estimated at USD 60 billion (WHO, 2002). In China, herbal medicine played a significant role in containing and treating the 2004 severe acute respiratory syndrome (SARS) epidemic [23]. Additionally, the Generally Recognized-as-Safe (GRAS) list, approved by the Food and Drug Administration (FDA) and Environmental Protection Agency (EPA) in the USA, includes many of the commercial-plant-derived essential oils [24]. Apart from being a key ingredient in traditional cuisine and a fragrance component, *D. carota*, or ‘carrot’, has been used since ancient times for its nutritional and medicinal properties. Ancient Greeks and Romans valued and mentioned the plant in various reputable writings, and the different parts of the carrot were popularly utilized as medicinal sources in the treatment of many diseases [25]. Although less common than the domesticated variant of the carrot, the wild carrot has been used in foods as a flavoring agent in baked goods, frozen dairy, and meat products, as well as in gelatins, soups, and relishes [26].

The lesser-known subspecies, wild carrots, are no less important subtypes and have been utilized for various indications. It was first described in the third century BCE by the ancient Roman Diphilus of Siphnos to possess diuretic properties and by Pliny the Elder for its aphrodisiac effects [27]. According to “The Encyclopedia of Medicinal Plants”, the wild carrot is a very beneficial plant that possesses hepatoprotective, diuretic, and detoxifying properties [28]. An infusion of its leaves prevents kidney-stone formation, treats existing stones, combats cystitis, promotes the pituitary gland release of gonadotrophins, and treats parasitic infections [12,28]. It presents itself as an effective remedy for numerous digestive, kidney, bladder, and menstrual ailments, as well as dropsy, flatulence, and edema [5,11,12,13,29,30,31,32]. The Romans were known to use the wild carrot as a component for contraception, a potent emmenagogue and an inhibitor of implantation [33]. Additionally, the plant was known as the traditional “morning after” contraception through the stimulation of the uterus [12,33,34]. Moreover, Duke et al. reported that the wild carrot exhibits carminative, diuretic, emmenagogue, and anthelmintic properties [35]. The seed oil has also been used in anti-wrinkle creams [36]. In European and Middle Eastern folk medicine, wild carrot oil is used as an antiseptic and anti-inflammatory therapy for prostatitis and cystitis [5,11,12,13], diabetes mellitus, and gastric ulcers, as well as for myorelaxation [37].

Wild carrot is consumed in salads as part of the Mediterranean diet and is used as a food additive in some recipes [38]. People eat its young taproot cooked, consume its flower umbels after French frying, and use its seed oil as a flavoring agent in beverages and food products [39]. In Lebanese folk medicine, the plant is used for protection against hepatic diseases and the treatment of diabetes, gastric ulcers, muscle pain, and cancer. Professor Nehme, in his book the ‘Wild Flowers of Lebanon’, mentioned that the aromatic seeds of the wild carrot were used as a vermifuge, a diuretic, an antidote for snake bites, and for sterility [40]. Traditional medicine presents *D. carota*, whether cultivated or wild, as a highly beneficial plant with promising potential for treating various ailments. The traditional uses of the wild carrot in various ethnic groups are presented in Table 1.

## 5. Phytochemistry and Bioactive Compounds/Chemical Composition

To better understand and characterize the medicinal benefits of the wild carrot, a phytochemical investigation is warranted. Numerous scholars have conducted extensive studies on the different *D. carota* subspecies to identify and characterize their bioactive constituents. As illustrated in Table 2, terpenoids and phenolics emerge as two significant chemical classes identified, with the terpenoids further classified into monoterpenes (e.g., α-pinene and geranyl acetate), sesquiterpenes (e.g., humulene and carotol), diterpenes (e.g., phytol), triterpenes (e.g., squalene), and tetraterpenes (e.g., α-carotene). Phenolics encompass phenylpropanoids, flavonoids, and tannins.

As is evident in Table 3, *D. carota* subspecies have proven to be rich in various chemical components, with some being familiar and well-studied while others are new and worth further attention. In particular, the monoterpenes α-pinene, geranyl acetate, and sabinene, along with the sesquiterpene carotol, stand out as the most prominent constituents in different *D. carota* subspecies, exhibiting variable percentages. These compounds are noteworthy for their diverse pharmacological properties, including anti-inflammatory, antibacterial, and antifungal activities [51,52,53,54,55,56,57,58,59,60,61,62,63]. Although the country of origin and the plant organ differ, most of the components are shared among all subspecies, with some exceptions that are clearly evident. For example, β-2-himachalen-6-ol is a novel sesquiterpene that was extracted, identified, and reported for the first time by a Lebanese team of researchers while studying the Lebanese variant of *D. carota* ssp. *carota*. This compound is exclusive to this subspecies grown in Lebanon, as it was never reported in any other country while studying the same variant. Furthermore, the two phytochemicals, apiole and myristicin are unique to the Algerian *D. carota* ssp. *hispanicus*. Table 3 summarizes the different *D. carota* subspecies and their main chemical components.

## 6. Pharmacological Activities

The popular use of *D. carota* in traditional medicine has attracted the attention of many researchers to investigate the claimed benefits. Consequently, numerous studies have been carried out to investigate the proposed pharmacological properties. Relying on its ethnopharmacological indications, mainly to treat cancer, inflammation, and infections, both in vitro and, to a lesser extent, in vivo models were implemented to assess its pharmacological relevance. The findings of these studies are discussed in the sections below.

### 6.1. Anticancer Activity

With the increasing need for optimizing cancer therapy, extensive studies were conducted to assess the potential benefit of herbal options in oncology. Regarding the anticancer potential of wild *D. carota*, only *D. carota* ssp. *carota* was studied and demonstrated to possess these properties. Zeinab et al. [65] assessed the chemoprotective effects of Lebanese *Daucus carota* ssp. *carota* dried umbels acetone-methanol (50:50) oil extract on 7,12-dimethyl benz(a)anthracene (DMBA)-initiation-12-O-tetradecanoyl phorobol-13-acetate (TPA)-induced skin carcinogenesis in mice. The extract was administered through topical applications, gavage, and intraperitoneal (IP) injections. Both intraperitoneal and topical treatments displayed a significant inhibition of papilloma number and volume, accompanied by an improved tissue histology.

Moreover, recent findings revealed that *D. carota* oil extract (DCOE) demonstrated substantial preventive and therapeutic effects against DMBA-induced breast cancer in rats. Animals pre-treated with DCOE exhibited prolonged survival and reduced tumor incidence. Also, DCOE treatment post-tumor induction resulted in a significant inhibition of tumor volume [80]. Shebaby et al. [37] reported that DCOE treatment exhibited important cytotoxicity against several human cancer cell lines (colon: HT-29 and Caco-2; breast: MCF-7 and MDA-MB-231). According to the authors, the anticancer activity of DCOE may be attributed to the presence of major sesquiterpenes including β-caryophyllene, caryophyllene oxide, α-humulene, and a prominent compound C_15_H_26_O (later identified as β-2-himachalen-6-ol), in addition to other minor constituents acting in synergy. Similarly, Ambrož et al. [81] reported that β-caryophyllene and its oxide, as well as α-humulene, obtained from *Myrica rubra* leaves, exhibited significant anticancer properties in CaCo-2 cells and other intestinal cancer cell lines, synergistically potentiating the efficacy of doxorubicin cytotoxicity. Additionally, in 2007, Legault and Pichette [82] showed that treatment with β-caryophyllene caused a significant enhancement of the anticancer effects of isocaryophyllene, α-humulene, and paclitaxel on MCF-7 cells. Further studies also confirmed the cytotoxic activity of these highlighted compounds [83,84,85,86,87]. In 2019, Hammami et al. [79] reported that while (4R)-1-*p*-menthen-6,8-diol, a polyol menthane monoterpenoid, isolated from *D. carota* ssp. *hispidus*, did not induce cytotoxicity against human mouth squamous carcinoma (HSC-2) and human cervical (HeLa) cancer cells, while (1R,2R,4R)-*p*-menthane-1,2,4-triol and (4R)-1-*p*-menthen-4,7-diol (2) exhibited weak activity against the same cell lines.

Furthermore, in 2015, Tawil et al. [25] reported that DCOE induced caspase-dependent apoptotic cell death in human Acute Myeloid Leukemia (AML) cells, partially through the Mitogen-Activated Protein Kinase (MAPK)-dependent mechanism. These kinases regulate important cellular mechanisms such as proliferation, stress responses, immune defense, and apoptosis [88]. DCOE was eluted into four fractions (pentane:diethyl ether, 50:50 (F2), diethyl ether, 100%) and F4 (chloroform:methanol, 93:7) and tested against various human cell lines [89]. The pentane (100%) and pentane/diethyl ether (50:50) fractions displayed the highest anticancer activity against MDA-MB-231, MCF-7, HT-29, and Caco-2 cells. Inhibition of cell proliferation in MDA-MB-231 and HT-29 cells was attributed to cell cycle arrest and apoptosis. The latter was mediated via the suppression of the MAPK/Erk pathway in MDA-MB-231 cells and the PI3K/Akt and MAPK/Erk pathways in HT-29 cells [89,90]. In another study, the four DCOE fractions exhibited significant in vitro cytotoxic effects against both tumorigenic and non-tumorigenic human epidermal keratinocytes (HaCaT cells and HaCaT-ras variants). The anti-tumor effect of the pentane:diethyl ether (50:50) fraction was also observed in a DMBA/TPA skin carcinogenesis mouse model, where a significant decrease in papilloma incidence, yield, and volume was noted at weeks 15, 18, and 21. This activity was attributed to its major compound β-2-himachalene-6-ol [91]. Furthermore, the same fraction displayed a significant decrease in cell motility and invasion as well as an increase in cell adhesion in lung (A549), skin melanoma (B16F-10), breast (MDA-MB231), and SF-268 (glioblastoma) cells. These effects are partially due to the inhibition of ρ-GTPases, Rac, and CDC42 [92]. Later on, β-2-himachalen-6-ol was reported to cause significant cytotoxicity against various human cancer cell lines by inducing apoptosis and inhibiting the PI3K/Akt and the MAPK/Erk pathways. Moreover, β-2-Himachalen-6-ol decreased 2D cell motility and 3D invasion and increased cell adhesion in SF-268 [64]. Flow cytometry assays demonstrated that the treatment of HaCaT-ras II-4 with β-2-himachalen-6-ol caused a cell cycle arrest, an up-regulation of the p21 protein, and induced apoptosis through increasing the levels of cleaved caspase-3 and pro-apoptotic BAX proteins and by decreasing in the anti-apoptotic Bcl-2 protein [93,94]. Furthermore, β-2-himachalen-6-ol displayed significant chemo-preventive and chemo-therapeutic effects in the DMBA/TPA skin cancer mouse model by reducing papilloma incidence, yield, and volume. This effect was partially attributed to promoting the apoptosis and inhibition of the MAPK/ERK and PI3K/AKT pathways, without inducing significant toxicity [93,95]. Additional in vitro studies showed that β-2-himachalen-6-ol caused cell cycle arrests and induced apoptosis and DNA fragmentation in SW1116 colon cancer cells. A Western blot analysis showed that β-2-himachalen-6-ol upregulated the p53, p21, and BAX proteins while downregulating the Bcl-2, procaspase-3, and PARP proteins. In 2019, Daaboul et al. [96] revealed that β-2-himachalen-6-ol treatment caused a dose-dependent decrease in cytotoxicity, induced cell cycle arrests, and promoted apoptosis in murine breast cancer cell line 4T1, along with a reduction in the primary tumor size and an inhibition of metastasis in mice. Thus, the aforementioned literature provides enough evidence to support the claimed traditional anticancer use of *D. carota* ssp. *carota* in Lebanon. In conclusion, we believe that *D. carota* ssp. *carota* is worth further pre-clinical investigations to confirm its efficacy and safety. The above findings and Table 4 and Table 5 summarize the in vitro and in vivo anticancer effects of *D. carota* ssp. *carota*.

### 6.2. Antibacterial Activity

With the increasing resistance to antibacterial agents and the need for alternatives to the current options, few studies have assessed the role of *D. carota* in eradicating bacterial infections. Only *D. carota* ssp. *carota*, *hispanicus*, *maritimus*, and *maximus* have been investigated and have been shown to possess antibacterial properties against various bacterial species. In 2005, Staniszewska et al. [8] tested the essential oil of *D. carota* ssp. *carota* umbels from Poland against four different species of bacteria, revealing a greater efficacy against Gram-positive bacteria (*Staphylococcus aureus* and *Bacillus subtilis*) than Gram-negative bacteria (*Pseudomonas aeruginosa* and *Escherichia coli*). Moreover, *C. jejuni* (reference strain and clinical isolates F38O11, LV7, and LV9) as well as other bacterial strains of the *Campylobacter* genus (*C. coli* and *C. lari*) were found to be equally susceptible to the antibacterial effect of the aerial parts of *D. carota* ssp. *carota* harvested in France [5]. Additionally, the oil was tested against a multidrug-resistant *C. jejuni* strain (99T403), and it exhibited comparable growth inhibition to that of non-resistant strains. The antibacterial effect was attributed to phenylpropanoids, elemicin, and (*E*)-methylisoeugenol. Ripe and unripe fruits, flowers, roots, leaves, and stems of *D. carota* ssp. *carota* oil extracts from Serbia were also investigated [69]. While the most remarkable activity was observed against *Bacillus cereus*, the highest inhibition was attained by the ripe and unripe fruits and the lowest with the flower oil. The tested oils were more active than Streptomycin against *Escherichia coli* and *Pseudomonas aeruginosa*. Essential oils of *D. carota* ssp. *carota* ripe umbels from Portugal was also tested against Gram-positive (*Staphylococcus aureus*, *Bacillus subtilis*, and *Listeria monocytogenes*), and Gram-negative (*Salmonella typhimurium* and *Escherichia coli*) strains, revealing its higher effectiveness against Gram-positive bacteria [66]. The observed inhibition was attributed to the major compounds α-pinene and geranyl acetate. Several studies have highlighted the antibacterial potential of α-pinene against numerous bacterial species. Kovac et al. [38] reported that α-pinene modulates *Campylobacter jejuni* antibiotic resistance by reducing the MIC value of triclosan, erythromycin, and ciprofloxacin by up to 512 times. This is achieved through targeting the antimicrobial efflux systems, enhancing microbial influx, reducing membrane integrity, and disrupting metabolisms [51]. Also, α-pinene, isolated from Korean citrus species, demonstrated significant antibacterial activity against *Staphylococcus epidermidis* and *Propionibacterium acnes* [53]. Another study reported that α-pinene demonstrated activity against the *Staphylococcus aureus*, *Staphylococcu epidermidis*, *Streptococcus pneumoniae*, and *Streptococcus pyogenes* strains [60]. The antibacterial activity of α-pinene against various bacterial strains has been consistently reported in multiple studies [51,52,53,54,55,56,57,58,59,60,69].

Asilbekova et al. [71] reported that essential oils of different parts (leaves, flowers, petals, fruits) of *Daucus carota* ssp. *carota* from Uzbekistan demonstrated modest activity against *Staphylococcus aureus* and *Bacillus subtilis*. Furthermore, in 2009, Wehbe et al. [97] showed that the aqueous and methanolic extracts of Lebanese *D. carota* ssp. *carota* umbels exhibited a minimal inhibition of *Staphylococcus aureus* meti S and meti R (MIC20 and 10 mg/mL, respectively), while no effect was observed against Gram-negative bacteria. Essential oils extracted from flowers and roots of Tunisian *D. carota* ssp. maritimus were also tested, showing broad-spectrum antibacterial activity against *P. aeruginosa*, *E. coli*, *E. faecalis*, *K. pneumoniae*, and *S. typhimurium* [76]. The flower-derived oil was more effective against *E. coli* (ESLβ), while the root-derived oil demonstrated greater efficacy against *S. aureus*, *S. pneumonia*, *Shigella* spp., and *E. faecalis*. The authors attributed these pharmacological activities to the significant presence of sabinene and terpinen-4-ol, as well as phenolic compounds like dillapiole and myristicin.

Sabinene has been reported to possess antibacterial activities against several Gram-positive and Gram-negative bacteria. The sabinene-type of the essential oil obtained from berries and leaves of *Juniperus excelsa* from Macedonia exhibited moderate activity against *Streptococcus pyogenes*, *Haemopilus influenzae*, *Campylobacter jejuni*, and *Escherichia coli* [98]. Moreover, sabinene, a major component of the *Zornia diphylla* essential oil (43.1%, whole plant), showed promising antibacterial activities, especially against in vitro *Salmonella typhi* [99].

Algerian *D. carota* ssp. *hispanicus* roots’ and aerial parts’ essential oils were shown to exhibit varied growth-inhibitory effects against Gram-positive and Gram-negative bacteria. The authors suggest that the antibacterial activity could be attributed to phenolic compounds such as myristicin and apiole [77]. Additionally, the seed oil obtained from the Italian *D. carota* ssp. *maximus* demonstrated significant inhibitory activity against most of the tested Gram-positive strains, particularly Staphylococcus, and only two Gram-negative pathogens, *Acinetobacter* and *Stenotrophomonas maltophilia* ICE272 [75]. Furthermore, the oil extract of Tunisian *D. carota* ssp. *carota* umbels displayed antibacterial activity against *E. coli* ATCC 35218 and *S. aureus* ATCC 43300 [9].

The above findings and Table 6 summarize the antibacterial activity of *D. carota* ssp. *carota*, *hispanicus*, *maritimus*, and *maximus*. The reported MIC values indicate weak to moderate antibacterial activity, which does not strongly support the claims of *D. carota* being an effective antiseptic and a remedy for prostatitis and cystitis in traditional medicine [5,11,12,13]. However, further in vitro and in vivo studies are recommended to confirm such claims.

### 6.3. Antifungal Activity

The antifungal properties of *D. carota* subspecies were studied against different fungal strains. *D. carota* ssp. *carota*, *gummifer*, *halophilus*, *hispanicus*, and *maximus* were found to exhibit significant antifungal activities. Oil extract derived from different plant organs of *D. carota* ssp. *carota* demonstrated both fungistatic and fungicidal activities, with *Fulvia fulvum* being the most susceptible and *Trichoderma viride* and *Aspergillus ochraceus* being the most resistant. Among the plant organs, the oil from unripe fruits exhibited the strongest antifungal activity, followed by ripe fruit oil, root oil, stem oil, leaf oil, and flower oil [69]. In 2017, Asilbekova et al. [71] showed that fruit essential oil of *D. carota* ssp. *carota* from Uzbekistan showed antifungal activity against *C. albicans*. However, the essential oils of *D. carota* ssp. *carota* umbels from Poland displayed weak inhibitory activity against *Candida albicans* and *Penicillium expansum* [8]. Furthermore, the antifungal activity of *D. carota* ssp. *carota* blooming and flowering umbels’ essential oils from Sardinia and Portugal as well as ripe umbels from Portugal were investigated against yeast, dermatophytes, and filamentous fungi [6,66]. The oils demonstrated greater antifungal potential against *Cryptococcus neoformans* and dermatophyte strains compared to *Candida* and *Aspergillus* strains. The mechanism of action involves an interference with preformed biofilms, reducing the amount of attached biomass and inhibiting germ-tube formation. According to the authors, the antifungal effect was attributed to the presence of α-pinene, geranyl acetate, and α-limonene [66]. Previous reports indicated that α-pinene displayed antifungal activity against several *Candida albicans* strains [54] as well as other fungal species such as *C. neoformans* and *R. oryzae* [59,100]. The antifungal activity was mainly attributed to an interference with biofilm formation. α-limonene has also been shown to possess significant antifungal properties against different fungal strains, such as *Sclerotinia sclerotiorum*, *Zygosaccharomyces rouxii*, *Trichophyton rubrum*, Candida, and yeast [101,102,103,104,105,106]. Moreover, geranyl acetate was found to have a broad antifungal spectrum against different dermatophyte strains, *Crytococcus neoformans*, and *Candida guillermondii* [107].

The antifungal properties of other *D. carota* subspecies were also investigated. In 2015, Valente et al. [24] revealed that an essential oil from aerial parts of *D. carota* ssp. *gummifer* was found to be more active against *C. guillermondii*, dermatophyte strains, and *C. neoformans* compared to other Candida strains. The authors attributed the antifungal properties to the high contents of geranyl acetate and α-pinene. The essential oil from aerial parts of Portuguese *D. carota* ssp. *maximus*, also tested for antifungal activity, demonstrated a broad antifungal spectrum against many fungal strains, especially Aspergillus, Candida, dermatophytes and *Cryptococus neoformans* [74]. The oil also exhibited a higher antifungal activity against Aspergillus and Candida strains compared to other Portuguese subspecies and various *D. carota* ssp. *taxa* from different countries [8,69]. Similarly, the authors named α-pinene and geranyl acetate as the main contributors to these antifungal effects. In another study by Tavares et al. [70], essential oils from umbels of *D. carota* ssp. *halophilus* were shown to possess significant antifungal activity against dermatophyte strains, linked to the presence of elemicin. In addition, an essential oil extracted from roots of *D. carota* ssp. *hispanicus* was found to display a stronger antifungal effect against *A. flavus* than an oil extracted from the aerial parts, and these effects were ascribed to the presence of the phenolic derivatives myristicin and apiole [77].

These findings highlight the moderate broad antifungal properties of *D. carota* ssp. *carota*, *hispanicus*, *maritimus*, and *maximus*. As such, the oil of these various *D. carota* spp. may be recommended as an antifungal remedy that could be used in traditional medicine, as no previous claims of such activity have been reported. Further studies are required to confirm these findings using in vitro and in vivo models. It is also important to identify the active ingredients responsible for this antifungal property and to elucidate the mechanism of action involved. Table 7 summarizes the antifungal effects of *D. carota* ssp. *carota*, *gummifer*, *halophilus*, *hispanicus*, and *maximus.*

### 6.4. Antioxidant Activity

Free radicals are known to contribute to several ailments, including cardiovascular, cancer, and degenerative diseases. Numerous antioxidant studies have been conducted on *D. carota* ssp. *carota*, with only one study on *D. carota* ssp. *gummifer*. Akgul et al. [108] reported that oil extracted from *D. carota* L. ssp. *carota* flowers from Turkey possesses significant DPPH radical scavenging properties, and this effect was stronger than that of individual compounds extracted from the oil. Similarly, Shebaby et al. [37] indicated antioxidant properties for the Lebanese *D. carota* ssp. *carota* umbel oil extract (DCOE) using the FRAP and DPPH assays. The results exhibited significant radical scavenging activity of DCOE in comparison to Trolox. The authors attributed the scavenging role of DCOE to the high levels of terpenes, phenols, and polyphenolic compounds. Additionally, an essential oil from the Algerian wild carrot flower demonstrated the strongest antioxidant activity compared to the aerial part essential oils before and after flowering. The antioxidant capacity of the oils was due to the presence of monoterpenes such as α-pinene, β-bisabolene, sabinene, and limonene [67]. Another study by Ksouri et al. [68] reported that the methanolic and essential oil of Algerian *D. carota* ssp. *carota* exhibit antioxidant activities in different in vitro assays, including DPPH and TBARS assays. However, the essential oil of *D. carota* ssp. *carota* ripe umbels from Portugal exhibited weak antioxidant effects using ABTS and peroxyl-induced oxidation inhibition assays [66]. On the other hand, a *D. carota* ssp. *gummifer* aerial parts’ essential oil displayed a substantial nitric oxide (NO) scavenging effect in all tested concentrations [24].

In vitro and in vivo antioxidant properties of DCOE fractions were also investigated [109]. The diethyl ether and chloroform/methanol fractions demonstrated the highest antioxidant activity in DPPH and FRAP assays. These effects could be attributed, at least in part, to the presence of phenolic compounds such as luteolin, kaempferol, apigenin, caffeic acid, and quercetin, all of which have been reported to display antioxidant properties [110,111,112,113,114].

The antioxidant properties of different fractions of DCOE were examined both in vitro and in living organisms in vivo [109]. The fractions made using diethyl ether and chloroform/methanol were found to have the strongest antioxidant activity when tested using the DPPH and FRAP assays. This activity is likely due to the presence of various phenolic compounds, including luteolin, kaempferol, apigenin, caffeic acid, and quercetin, which are known to have antioxidant properties [110,111,112,113,114]. On the other hand, the pentane fraction that is abundant in sesquiterpenes displayed moderate antioxidant activity when evaluated by DPPH assay. This can be linked to the presence of α-humelene and β-caryophyllene, which are known to have significant antioxidant effects [82,115,116,117]. Additionally, the sesquiterpene α-longipinene. found in pentane, diethyl ether, and chloroform/methanol fractions, was reported to exhibit antioxidant effects [118,119]. Furthermore, the pentane and pentane/diethyl ether fractions were highly effective at binding and neutralizing Fe^2+^ ions (strong chelating activity). Metal ions such as Fe^2+^ play a crucial role in the formation of hydroxyl radicals through the Fenton reaction, accelerating lipid peroxidation [120]. β-Caryophyllene, a prevalent sesquiterpene found in the pentane fraction, has been reported as a potent inhibitor of lipid peroxidation due to its ability to react with peroxyl radicals [121]. The antioxidant properties of DCOE fractions were also evaluated in vivo by measuring the activity of enzymes such as catalase (CAT), superoxide dismutase (SOD), and glutathione S-transferase (GST) in the livers of mice intoxicated with CCl_4_. The pentane fraction (200 mg/kg) and the chloroform/methanol fraction (50, 100, and 200 mg/kg) were found to increase CAT and SOD activity; however, the pentane/diethyl ether and diethyl ether fractions were ineffective in restoring the enzymes’ activity to normal levels [109]. All these studies describe the multimodal mechanism of action of *D. carota* in combating free radicals, highlighting its role as a prominent antioxidant agent.

### 6.5. Anti-Inflammatory Activity

Wild carrot has traditionally been used in folk medicine to treat various inflammatory conditions like cystitis and prostatitis. In light of this, researchers have conducted numerous studies to examine the anti-inflammatory properties of various *D. carota* ssp. including spp. *carota*, *gummifer*, and *maximus*. Both the aqueous (100, 200, and 400 mg/kg BW) and methanolic (70, 140, and 280 mg/kg BW) extracts of the Lebanese *D. carota* ssp. *carota* umbels displayed a significant reduction in acute and chronic inflammation in rats. The anti-inflammatory effects observed were similar to those of the non-steroidal anti-inflammatory drug (NSAID) diclofenac and were associated with the presence of terpenes and flavonoids in the plant extracts [97]. Additionally, research has found that an essential oil extracted from the aerial parts of *D. carota* ssp. *gummifer* possesses potent anti-inflammatory properties. It was able to inhibit the production of nitric oxide (NO) in macrophages and microglia cultures that were stimulated with lipopolysaccharide (LPS) [24]. Similar studies have also been conducted to investigate the effects of an essential oil from the aerial parts of *D. carota* ssp. *maximus* on the NO production in LPS-stimulated macrophages and microglia cells. At a non-toxic concentration (0.32 μL/mL), the essential oil was able to decrease NO production by 20.7% and 35.8% in macrophages and microglia cells, respectively. The anti-inflammatory properties of the oil were partly attributed to the presence of high levels of geranyl acetate and α-pinene [74]. Multiple studies have also confirmed the anti-inflammatory effects of these two terpenes. In particular, (+)-α-pinene was found to be effective in inhibiting the IL-1β-stimulated inflammatory and catabolic pathways, thus exhibiting potential anti-osteoarthritic activity [122]. It also reduced IL-6 and TNF-α formation in macrophages of rats and decreased nitrite production [123]. On the other hand, geranyl acetate showed significant anti-inflammatory activity in vitro, without affecting macrophages’ and keratinocytes’ viability [107]. Despite the relatively high doses used in the above studies, the findings align with the traditional use of wild carrot as an anti-inflammatory remedy, especially considering the use of a crude extract. Additional in vivo studies can further support its traditional use.

### 6.6. Miscellaneous

In a study by Wehbe et al. [97], the researchers examined the ability of aqueous and methanolic extracts of Lebanese *D. carota* ssp. *carota* umbels to protect against ethanol-induced gastric damage in rats. Both extracts demonstrated a protective effect, with the methanolic extract showing a higher curative ratio compared to the aqueous extract and the group treated with cimetidine. According to these authors, this gastro-protective effect is likely due to the presence of flavonoids and tannins in the plant. Additionally, the aqueous extract was found to decrease levels of HDL, without having any significant impact on total cholesterol, triglycerides, and LDL concentrations. Furthermore, no significant effects were observed on the glucose, insulin, and amylase activity in the blood after the use of the aqueous extract. In another study by Muturi et al. [72], it was revealed that the essential oil of *Daucus carota* ssp. *carota* displayed significant toxicity against mosquito larvae, suggesting the potential use of wild carrot essential oils as a bio-pesticide. The contraceptive effects of the wild carrot have been investigated as well. Several in vivo studies have shown that extracts from wild carrot seeds can impede pregnancy through diverse mechanisms, including acting as anti-zygotic and/or blastocystotoxic agents [124] and inhibiting implantation through anti-progestogenic and antioestrogenic effects [125,126,127].

## 7. Safety/Toxicological Evaluation

In addition to the assessment of their efficacy, subspecies *carota*, *gummifer*, *halophilus*, and *maximus* were evaluated for their safety. The aqueous extract of Lebanese *D. carota* ssp. *carota* umbels showed no significant variations in liver enzyme concentrations (SGOT, SGPT, ALP, and LDH) in rats, suggesting that the extract can maintain the structural integrity of the hepatocellular membrane [97]. Tawil et al. [25] revealed an extremely low susceptibility of normal human peripheral blood mononuclear cells (PBMCs) to DCOE methanol:acetone (1:1) treatment. Additionally, according to Dixon’s up and down model, the LD_50_ of the major compound β-2-himachalen-6-ol (HC), obtained from Lebanese *D. carota* ssp. *carota* umbels, was 1000 folds higher than that of cisplatin, highlighting the low toxicity profile of DCOE in adult Balb/c mice [64]. In another study, chronic treatments with HC displayed mild hepatotoxicity with no adverse effects on the kidneys of the treated mice. The HC-treated groups showed the highest survival rate among all groups [95]. Furthermore, a study using a 1,2-dimethylhydrazine (DMH) colon carcinogenesis model in six black mice found that HC did not cause significant liver toxicity at doses of 10 and 25 mg/kg, while moderate hepatotoxicity was observed at a concentration of 50 mg/kg [94]. Daaboul et al. [93,96] also reported that no body weight changes were observed after HC treatment in contrast to cisplatin, which exhibited significant weight reductions in the animals. No variations in liver enzyme profiles were reported after HC treatment, indicating no hepatotoxicity. On the other hand, the essential oil of *D. carota* ssp. *carota* derived from ripe umbels from Portugal was tested on keratinocytes, alveolar epithelial cells, macrophages, and hepatocytes [66]. The oil was considered safe at concentrations below 0.64 μL/mL, with macrophages being the least vulnerable (92.83% ± 1.04 cell viability) and hepatocytes the most susceptible (60.73% ± 6.51 cell viability).

A previous report indicated that an essential oil derived from the aerial parts of *D. carota* ssp. *gummifer* does not affect the cell viability of macrophages at concentrations below 1.25 μL/mL [24]. Comparable results were observed for liver cells (hepatocytes) and skin cells (keratinocytes). However, it was somewhat more toxic to microglia cells, leading to decreased cell viability when used at concentrations above 0.64 μL/mL [24].

When testing *D. carota* ssp. *halophilus* umbel essential oils at concentrations possessing significant antifungal activity, no cytotoxicity was detected in mouse skin dendritic cells [70]. Lastly, the safety of the essential oil from aerial parts of Portuguese *D. carota* ssp. *maximus* was assessed on human keratinocytes and hepatocytes as well as mouse macrophages and microglial cells [74]. The results showed that cell viability at concentrations lower than 1.25 μL/mL was not affected for all cell lines except microglial cells, where cytotoxicity was observed at a concentration as low as 0.64 μL/mL.

## 8. Discussion and Conclusions

Traditional medicine has always been the backbone of drug discovery, and with the accelerating need for new and improved medication options, an in-depth study of the pharmacological potential of natural remedies is essential. As previously reported, the wild carrot has a long history of traditional use for treating a variety of conditions, including pain, stomach ulcers, diabetes, cancer, inflammation, and infections [5,11,12,13].

To better characterize the therapeutic potential of wild *D. carota*, it is essential to investigate the interplay between its traditional uses and pharmacological activities in relation to its chemical composition. As previously cited in this review, *D. carota* ssp. *carota* has a longstanding traditional use in Lebanon to fight oncological ailments, a usage substantiated by numerous studies underscoring its significant anticancer properties, attributed mainly to the presence of the distinctive compound β-2-himachalen-6-ol [25,37,64,65,89,90,91,92,93,94,95,96]. The wild carrot has also been traditionally employed by the Lebanese and Persian communities in the treatment of gastric ulcers [40,43], as supported by a study indicating that both aqueous and methanolic extracts possess protective effects against ethanol-induced gastric damage in rats. The authors attribute this gastro-protective activity to the presence of flavonoids and tannins within the plant [97]. Furthermore, in another investigation, the plant extract exhibited hepatoprotective effects in the livers of CCl_4_-intoxicated mice by increasing or restoring the normal levels of the antioxidant liver enzymes [109]. These findings offer additional support for the use of wild carrots in traditional folk medicine for the protection against hepatic diseases [40].

Regarding the antimicrobial activities, various subspecies, including *carota*, *hispanicus*, *maritimus*, and *maximus*, have exhibited mild to moderate antibacterial properties against various bacterial species, aligning with their historical use in folk medicine as antiseptics for treating bacterial infections, such as prostatitis and cystitis [5,11,12,13]. Notably, α-pinene and geranyl acetate emerged as key constituents contributing to this antibacterial efficacy, which is in line with other studies that emphasize the antibacterial potential of α-pinene [51,52,53,54,55,56,57,58,59,60]. While folk medicine does not explicitly mention the use of the wild carrot for fungal infections, multiple studies underscore its significant antifungal activities against diverse fungal strains, particularly within *D. carota* ssp. *carota*, *gummifer*, *halophilus*, *hispanicus*, and *maximus* [6,8,24,66,69,70,74,77]. The phytochemicals responsible for this antifungal activity, namely, α-pinene, geranyl acetate, and α-limonene, align with findings from related studies [54,100,101,102,103,104,105,106,107]. In addressing inflammatory conditions, *D. carota* ssp. *carota*, *gummifer* and *maximus* have demonstrated noteworthy anti-inflammatory properties [24,74,97] that were mainly attributed to the presence of α-pinene and geranyl acetate [107,122,123]. This aligns with the historical utilization of wild carrots in folk medicine for the treatment different inflammatory disorders like pain, prostatitis, and cystitis [5,11,12,13]. The observed anti-inflammatory effect of the wild carrot may also be attributed to the existence of antioxidant compounds in the plant extract [24,37,66,67,108,109]. Notably, various terpenes including α-pinene, β-Caryophyllene, β-bisabolene, sabinene, limonene, and α-longipinene along with polyphenols like luteolin, kaempferol, apigenin, caffeic acid, and quercetin are recognized for their radical scavenging activities [110,111,112,113,114,118,119,121]. An elevated production of reactive oxygen species (ROS) is linked to oxidative stress and the oxidation of proteins [128]. This, in turn, triggers inflammatory mediators and numerous inflammatory signals in response to protein oxidations [129]. Moreover, an excess of ROS can induce tissue injury, activating the inflammatory process [130,131]. Hence, the well-established antioxidant properties of the diverse terpenes and polyphenols present in the wild carrot play a crucial role in their anti-inflammatory effects by disrupting the cycle of ROS-induced inflammation. These findings reinforce the convergence of scientific research with traditional practices, elucidating the specific entities responsible for desired effects. Other pharmacological properties of the wild carrot also confirm its practice in traditional medicine. Based on several in vivo studies, various mechanisms may contribute to the anti-fertility effect of the wild carrot, including an impact on the estrous cycle and anti-progestogenic activity [125,126,127]. These results further emphasize the ethnopharmacological practices of the wild carrot, both as a contraceptive agent and in its capacity as an emmenagogue [27,33,34].

Critically reviewing the literature available on the wild carrot reveals few limitations in current studies. To begin with, searching for *D. carota* in the literature revealed that authors do not always specify the subspecies that they refer to in their studies, which is an essential piece of information due to intraspecific polymorphism of these species. Usually, when referring to “*D. carota*”, authors target one single *D. carota* ssp. *sativus*, the most known, cultivated, and edible subtype. Therefore, it is important to point out the confusion that overshadows the nomenclature of the carrot. A correct naming and description of the subspecies must be employed by scholars for more accurate and credible research referencing. Another common mistake resides in not specifying the plant organ studied, namely, whether the authors used the whole plant or specific organs (leaves, roots, stems, flowers, etc.), and this creates confusion among readers. Moreover, α-pinene, geranyl acetate, β-2-himachalen-6-ol, and α-limonene are the main chemical constituents that are highlighted to exhibit significant pharmacological activities in their respective studies. However, other important components, such as myristicin, apiole, and dillapiole, have not been studied in this context, although they are present abundantly in different plant organs.

In conclusion, this review underscores the pharmacological merits of the wild carrot supported by its historical use in folk medicine and achieved through the synergistic interaction of specific essential chemical constituents. Nevertheless, certain applications outlined in traditional medicine remain unexplored. Therefore, there is a necessity to investigate the potential role of the wild carrot in conditions such as diuresis, toxicosis, kidney stones, bladder and gastric diseases, parasitic infections, and its potential as an antidote for snake bites. Furthermore, the current review specifies the chemical composition of various subspecies and identifies key phytochemicals responsible for their medicinal properties. *D. carota* ssp. *carota* from Lebanon emerges as the most studied, displaying unique anticancer activity. While almost all subspecies exhibit significant antibacterial, antifungal, antioxidant, and anti-inflammatory activities, further in vivo and pre-clinical studies are imperative to confirm these findings, ensuring the safety and efficacy of *D. carota* in diverse therapeutic applications. Moreover, the isolation and characterization of biologically active compounds and mechanistic elucidation should be prioritized for a more comprehensive understanding of the plant’s pharmacological properties.

## Figures and Tables

**Table 1 plants-13-00093-t001:** Traditional uses of the wild carrot in various ethnic groups.

Ethnic Groups	Traditional Uses	References
Native Americans	Used for various medicinal purposes, including as a diuretic and for digestive issues. Wild carrot seeds are included in a recipe specifically for female contraception.	[34,41,42]
Romans	Used as a component for contraception and contragestion, as well as for its diuretic properties and aphrodisiac effects.	[12,27,33,34]
Greeks	Seeds are recommended for their anti-fertility properties and their role as emmenagogues.	[33,34]
Persians	Leaf and root are recognized as diuretic, beneficial for fertility, and efficacious against conditions such as a cough, pleurisy, corrosive ulcers, and dropsy. Additionally, the seeds are noted for their diuretic and emmenagogic properties and is considered deobstruent and useful for alleviating intestinal pain.	[43]
Indians	Used in Ayurvedic medicine for various purposes, including as a diuretic and to treat digestive issues. The seeds are used as a method of controlling women’s fertility and the induction of abortion.	[44,45,46,47]
Turkish	Seeds employed in Turkish folk medicine for the treatment of gastrointestinal and respiratory disorders.	[48]
Chinese	Fruits utilized for the management of ascariasis, enterobiasis, and tapeworm disease. The roots and basal leaves are employed to strengthen the spleen and treat conditions such as dyspepsia and chronic dysentery.	[49,50]
Europeans	Wild carrot essential oil is used as an antiseptic for the treatment of cystitis and prostatitis as well to treat urinary calculus, gout, and lithuria.	[11,12,13]
Lebanese	Employed in the protection against hepatic diseases and in the treatment of diabetes, gastric ulcers, muscle pain, and cancer. Also, it is utilized as a vermifuge, a diuretic, an antidote for snake bites, and for managing sterility.	[40]

**Table 2 plants-13-00093-t002:** Classification of different classes of chemical compounds.

Class	Types	Examples
Terpenoids	Monoterpenes	Geraniol, Limonene, α-Pinene, β-Pinene, Sabinene, α-Terpinene, β-Myrcene, Geranyl acetate, Linalool, and α-Thujone
	Sesquiterpenes	Bergamotene, Humulene, Nerolidol, Selinene, Farnesol, Germacrene, Carotol, Caryophyllene, β-Himachalene, and β-Bisabolene
	Diterpenes	Phytol, Vitamin A_1_, and Cembrene
	Triterpenes	Squaline, Saponins, and Ginsenoide
	Tetraterpenes	Carotenoids (e.g., α-carotene, β-carotene, lycopene) and Xanthophylls (lutein)
Phenolics	Phenylpropanoids	Apigenin, Quercetin, Myristicin, and Methylisoeugenol
	Flavonoids	Luteolin, Apigenin, Quercetin, Myristicin, and Kaempferol
	Tannins	Gallic acid and Ellagic acid

**Table 3 plants-13-00093-t003:** Summary of the main chemical components of different *Daucus carota* subspecies from different countries of origin (percentages > 3% were reported).

*Daucus carota* ssp.	Plant Organ	Country	Main Components	References
*carota*	Umbels	Lebanon	β-2-himachalen-6-ol (33%), α-longipinene (3.22–15.87%), methyl linoleate (8.26%), (*E*)-methylisoeugenol (2.21–7.92%), 2-butanone (5.95%), α-Selinene (4.53–5.69%), Elemicin (4.03–4.93%), β-Asarone (4.07%), β Himachalene (2.24–4.63%), n-hexadecanoic acid (3.72%), humulene (3.27%), himachala-1,4-diene (3.09%), β-Bisabolene (1.76–3.78%)	[37,64,65]
	Flowering umbels	Portugal	α-Pinene (37.9%), geranyl acetate (15%), (*E*)-caryophyllene (4.9%), β-Pinene (3.5%),	[6]
	Umbels with ripe seeds		Geranyl acetate (65%), α-Pinene (13%)	
	Flowering umbels	Italy	Carotol (25.1%), 11αH-himachal-4-en-1-β-ol (21.6%), β-bisabolene (17.6%), elemicin (6.4%)	
	Umbels with ripe seeds		β-bisabolene (51%), (E)-methyl isoeugenol (10%), 11αH-himachal-4-en-1-β-ol (9%), elemicin (5.2%), α-longipinene (3.1%)	
	Ripe umbels	Portugal	Geranyl acetate (29%), α-Pinene (27.2%), Limonene (9%), 11αH-Himachal-4-en-1-β-ol (9.2%), Carotol (6.2%), β-Pinene (4.5%)	[66]
	Ripe umbels with mature seeds	Tunisia	Carotol (3.5–55.7%), Elemecin (1.4–35.3%), 11αH-Himachal-4-en-1-β-ol (12.7–17.4%), Sabinene (12–14.5%), α-Selinene (7.4–8.6%), Eudesm-7(11)-en-4-ol (8.2–8.5%), β-Bisabolene (5.5–7.6%), (*Z*)-β-Farnesene (1.6–5%), (*E*)-α-Bergamotene (0.2–3.8%)	[9]
	Herbs	Poland	Sabinene (30.1%), α-Pinene (30%), Terpinen-4-ol (6.1%), limonene (5.3%), myrcene (5.2%)	[8]
	Flowering Umbels		α-Pinene (42%), Sabinene (19.5%), limonene (3.7%), myrcene (3.1%)	
	Mature Umbels		Sabinene (40.5%), α-Pinene (17.2%), geranyl acetate (16.5%), Terpinen-4-ol (4.9%),	
	Aerial parts	France	(*E*)-methylisoeugenol (21.8–33%), β-Bisabolene (4.4–21.3%), Elemicin (11.4–16.3%), α-Pinene (15.9–24.9%), Sabinene (2.7–3.7%), Myrcene (2–3.5%), α-Terpinen-4-ol (0.5–3.5%)	[5,7]
	Flowers	Algeria	α-Pinene (10.9%), α-Asarone (9.8%), β-Bisabolene (7.6%), β-Caryophyllene (7.1%), Sabinene (7%), Daucol (3.2%), Limonene (3%)	[67]
	Leaves + Stems		α-Pinene (10.6%), α-Asarone (9.4%), β-Bisabolene (9.3%), Sabinene (7.2%), Carotol (6.8%), E-α-Bisabolene (6.3%), Daucol (5.3%), β-Caryophyllene (4.3%), Limonene (4%)	
	Aerial parts		α-Pinene (21.3%), α-Asarone (18.4%), β-Bisabolene (7.3%), Sabinene (6.5%), Limonene (6.4%), Carotol (3.5%), Terpinen-4-ol (3.5%), β-Caryophyllene (3.3%), E-α-Bisabolene (3.2%)	
	Leaves		α-Pinene (27.44%), sabinene (25.34%), Germacrene D (16.33%)	[68]
	Seeds		Geranyl acetate (52.45%), Cedrone S (14.04%), Asarone (11.39%), β-bisobolene (4.83%), Ar-himachalene (3.54%)	
	Ripe fruits	Serbia	Sabinene (27.16%), α-pinene (21.3%), α-muurolene (8.23%), β-caryophyllene (6.82%), α-ylangene (5.21%), β-Pinene (3.9%)	[69]
	Unripe fruits		α-muurolene (10.97%), sabinene (10.67%), caryophyllene oxide (7.7%), α-amorphene (7.57%), α-pinene (7.05%), carotol (6.15%), dimenone (5.28%), α-ylangene (4.88%),	
	Flowers		α-Pinene (51.23%), Limonene (9.59%), Sabinene (8.62%), β-Myrcene (7.18%), Terpinen-4-ol (3.48%), β-Pinene (3.35%)	
	Roots		Sabinene (36.39%), α-Pinene (24.56%), Limonene (6.53%), β-Pinene (5.39%)	
	Leaves		α-Pinene (30.83%), Limonene (8.6%), β-Myrcene (5.6%), Germacrene D (4.56%)	
	Stems		α-Pinene (18.53%), α-Bisabolol (6.02%), Limonene (5.74%), β-Myrcene (3.4%), Sabinene (3.23%)	
	Fruits	Portugal	Geranyl acetate (28.7–65%), α-Pinene (13–27.1%), 11αH-Himachal-4-en-1-β-ol (0.5–9.4%), Limonene (1.2–9%), β-Pinene (2.3–4.5%)	[70]
	Roots	Vienna	α-Terpinolene (26.2–56.3%), β-Pinene (4.1–8.2%), *p*-Cymene (2.7–7.4%), Sabinene (5.6–5.9%), γ-terpinene (0.9–5.6%), Limonene (5.5%), Myristicin (4.9–5.1%)	[1]
	Leaves		α-Pinene (20.9–44.8%), Sabinene (11.3–19.5%), Germacrene D (4.9–14%), Limonene (3.9–12.7%), Myrcene (4–11.2%), β-Pinene (1.3–5.9%), Caryophyllene (1.2–3.7%)	
	Fruits		Sabinene (21.5–46.6%), α-Pinene (23.5–30.4%), Geranyl acetate (3.9–28.1%), β-Pinene (3–13.1%), α-Thujene (1–8.8%), γ-terpinene (0.3–4.1%), Myrcene (3.4–3.9%)	
	Seeds	Lithuania	Sabinene (28.2–37.5%), α-Pinene (16–24.5%), Terpinen-4-ol (5–6%), γ-terpinene (2.9–6%), Limonene (3–4.2%)	[61]
	Leaves	Uzbekistan	Carotol (68.3%), Daucene (5%), *trans*-β-Farnesene (3.7%), β-Bisabolene (3.3%), α-Pinene (3.1%)	[71]
	Flowers		Carotol (68.8%), Daucene (4.7%), Daucol (3.4%), *trans*-β-Farnesene (3.3%)	
	Petals		Carotol (78.3%)	
	Fruits		Carotol (69.8%), Daucene (9%), *trans*-α-Bergamotene (4.7%), *trans*-β-Farnesene (3.7%)	
	Umbels	United States	α-Pinene (33.02%), β-Pinene (25.77%), Borneol (10.4%), Myrcene (6.41%), Limonene (5.34%), γ-terpinene (4.97%)	[72]
*maximus*	Ripe and mature fruits	Egypt	(*E*)-methylisoeugenol (37.22%), β-bisabolene (34.7%), β-Asarone (17.65%)	[2]
	Leaf		Preisocalamendiol (17.95%), Shyobunone (16.84%), β-Cubebene (12.72%), Tridecane (3.411%), Linalool (3.34%), (E)-2-Nonenal (3.22%)	
	Stem		Preisocalamendiol (32.69%), Shyobunone (24.33%), α-Pinene (4.37%), β-Cubebene (3.55%)	
	Fruits	Portugal	α-Pinene (10–25.9%), α-Asarone (5.8–25.8%), Geranyl acetate (3.4–16%), β-bisabolene (8.3–15.1%), (*E*)-methylisoeugenol (8.2–15.7%), Elemicin (4.9–13.6%), β-Pinene (4–6.8%), Limonene (1.8–3.3%) [73]	[70]
	Ripe umbels	Portugal	α-Pinene (22.2%), Geranyl acetate (16%), β-bisabolene (11.5%), α-asarone (9.8%), (*E*)-methylisoeugenol (8%), Elemicin (6%), β-Pinene (5.8%)	[74]
	Green seeds	Italy	Carotol (44.68%), β-bisabolene (12.72%), Isoelemicin (11.51%), Geranyl acetate (4.36%)	[75]
*maritimus*	Flowers	Tunisia	Sabinene (51.6%), Terpinen-4-ol (11%), *p*-Cymene (4.2%), Eudesm-6-en-4α-ol (3.6%)	[76]
	Roots		Dillapiole (46.6%), Myristicin (29.7%), Limonene (3.6%)	
*major*	Flowers	Italy	α-Pinene (24.4%), Sabinene (13.3%), Geranyl acetate (13%), *epi*-α-Cadinol (8.5%), Myrcene (4.8%), β-Oplopenone (4.3%)	[62]
	Fruits		Geranyl acetate (34.2%), α-Pinene (12.9%), Geraniol (6.9%), Myrcene (4.7%), epi-α-Bisabolol (4.5%), Sabinene (3.3%)	
*halophilus*	Flowering Umbels	Portugal	Sabinene (28.3–33.8%), α-Pinene (12.6–16%), Limonene (11–11.8%), (*E*)-methylisoeugenol (0.7–7.4%), Elemicin (5.9–6.2%), β-Bisabolene (0.4–5.3%), Terpinene-4-ol (4.1–4.8%), Myrcene (3.2–4.7%), β-Pinene (2.3–5.1%)	[73]
	Ripe Umbels		Elemicin (26–31%), Sabinene (27.6–29%), α-Pinene (10.1–12.2%), Limonene (5.5–6.5%), (*E*)-methylisoeugenol (0.5–6.9%)	
	Fruits		Elemicin (15–31%), Sabinene (9–29%), α-Pinene (12.2–23%), Limonene (5.5–12%), (*E*)-methylisoeugenol (0.5–7.4%), Terpinen-4-ol (2–4.7%)	[70]
*hispanicus*	Roots	Algeria	Apiole (80.3%), Myristicin (16.6%)	[77]
	Aerial parts		Myristicin (73.2%), Epiglobulol (5.1%), Germacrene D (3.1%)	
	Stems		Myristicin (66.9%), α-Thujene (4.3%)	
	Leaves		Myristicin (80.2%), Epiglobulol (3.1%)	
	Flowers		Myristicin (83.8%), Germacrene D (6.4%)	
*gummifer*	Fruits	Spain	Geranyl acetate (51.74–76.95%), Sabinene (4.42–11.13%), Terpinen-4-ol (0.93–8.17%), Linalool (3.97–5.18%) [78]	[78]
		Portugal	Geranyl acetate (18–55%), α-Pinene (11–31%), Carotol (5–15%), Sabinene (2.1–10%), Limonene (5.8–9%), Germacrene D (2–5.5%), β-Pinene (3.8–5.2%), Myrcene (2.1–3.7%)	[73]
	Ripe umbels	Portugal	Geranyl acetate (37%), Limonene (5.8%), α-Pinene (30.9%), β-Pinene (3.8%)	[24]
*hispidus*	Aerial parts	Tunisia	(4R)-1-*p*-menthen-6,8-diol, 1-*p*-menthen-4,7-diol, (1R,2R,4R)-*p*-menthane-1,2,4-triol, *β*-sitosterol 3-*O*-glucoside (abundance not specified) [79]	[79]

**Table 4 plants-13-00093-t004:** In vitro anticancer effects of Lebanese *D. carota* ssp. *carota*.

Extracts/Compounds	Detail	Concentration/Dose	References
Methanol/Acetone (1:1) *Daucus carota* oil extract (DCOE)	Cytotoxicty against U937	IC_50_ = 1 μg/mL	[25]
	Cytotoxicty against KG-1	IC_50_ = 1 μg/mL	
	Cytotoxicty against HL60	IC_50_ = 5.5 μg/mL	
	Cytotoxicty against TF1-VSrc	IC_50_ = 7.3 μg/mL	
	Cytotoxicty against Mono-Mac-6	IC_50_ = 12.4 μg/mL	
	Cytotoxicty against Mono-Mac-1	IC_50_ = 13.2 μg/mL	
	Cytotoxicty against TF1-VRaf	IC_50_ = 14.4 μg/mL	
	Cytotoxicty against MV-4-11	IC_50_ = 14.9 μg/mL	
	Cytotoxicty against ML1	IC_50_ = 19.6 μg/mL	
	Cytotoxicty against TF1-HaRas	IC_50_ = 26 μg/mL	
	Cytotoxicty against ML2	IC_50_ = 26.2 μg/mL	
	Cytotoxicity against human peripheral blood mononuclear cells (PBMCs)	IC_50_ > 100 μg/mL	
	Cytotoxicty against HT-29	IC_50_ = 34 μg/mL	[37]
	Cytotoxicty against MDA-MB231	IC_50_ = 27 μg/mL	
	Cytotoxicty against MCF-7	IC_50_ = 33 μg/mL	
	Cytotoxicty against Caco-2	IC_50_ = 30 μg/mL	
Pentane (100%) fraction (F1) Pentane: diethyl ether (50:50) fraction (F2) Diethyl ether (100%) fraction (F3) Chloroform: methanol (93:7) fraction (F4)	F1: Cytotoxicty against MDA-MB-231	IC_50_ = 17 μg/mL	[89]
	Cytotoxicty against MCF-7	IC_50_ = 22 μg/mL	
	F2: Cytotoxicty against MDA-MB-231	IC_50_ = 11 μg/mL	
	Cytotoxicty against MCF-7	IC_50_ = 32 μg/mL	
	F3: Cytotoxicty against MDA-MB-231	IC_50_ = 27 μg/mL	
	Cytotoxicty against MCF-7	IC_50_ = 48 μg/mL	
	F4: Cytotoxicty against MDA-MB-231	IC_50_ = 23 μg/mL	
	Cytotoxicty against MCF-7	IC_50_ = 43 μg/mL	
	F1: Cytotoxicty against HaCaT-ras A5	IC_50_ = 10.6 μg/mL	[91]
	Cytotoxicty against HaCaT-ras II4	IC_50_ = 10.2 μg/mL	
	Cytotoxicty against HaCaT	IC_50_ = 29.8 μg/mL	
	F2: Cytotoxicty against HaCaT-ras A5	IC_50_ = 14.6 μg/mL	
	Cytotoxicty against HaCaT-ras II4	IC_50_ = 11.4 μg/mL	
	Cytotoxicty against HaCaT	IC_50_ = 33.2 μg/mL	
	F3: Cytotoxicty against HaCaT-ras A5	IC_50_ = 19.1 μg/mL	
	Cytotoxicty against HaCaT-ras II4	IC_50_ = 17.8 μg/mL	
	Cytotoxicty against HaCaT	IC_50_ = 47.3 μg/mL	
	F4: Cytotoxicty against HaCaT-ras A5	IC_50_ = 16.2 μg/mL	
	Cytotoxicty against HaCaT-ras II4	IC_50_ = 14.5 μg/mL	
	Cytotoxicty against HaCaT	IC_50_ = 43.4 μg/mL	
Pentane (100%) fraction (F1) Pentane: diethyl ether (50:50) fraction (F2)	F1: Cytotoxicty against HT-29	IC_50_ = 22 μg/mL	[90]
	Cytotoxicty against Caco-2	IC_50_ = 18.5 μg/mL	
	F2: Cytotoxicty against HT-29	IC_50_ = 17.5 μg/mL	
	Cytotoxicty against Caco-2	IC_50_ = 19 μg/mL	
β-2-himachalen-6-ol (HC)	Cytotoxicty against B16F-10	IC_50_ = 13 μg/mL	[64]
	Cytotoxicty against Caco2	IC_50_ = 8 μg/mL	
	Cytotoxicty against MB-MDA-231	IC_50_ = 6 μg/mL	
	Cytotoxicty against A549	IC_50_ = 5 μg/mL	
	Cytotoxicty against SF-268	IC_50_ = 4 μg/mL	
	Cytotoxicty against SW116	IC_50_ = 18 and 14.5 μg/mL, 24 and 48 h respectively	[94]
	Cytotoxicty against HaCaT-ras II-4	IC_50_ = 7 μg/mL	[93]
	Cytotoxicty against HaCaT-ras II-4	IC_50_ = 8 μg/mL	[95]
	Cytotoxicty against 4T1	IC_50_ = 7 μg/mL	[96]

**Table 5 plants-13-00093-t005:** In vivo anticancer effects of Lebanese *D. carota* ssp. *carota*.

Cancer Model	Extracts/Compounds	Mode of Treatment (Dose and Frequency of Administration)	Results	References
DMBA/TPA-induced skin carcinogenesis model in mice	Methanol/Acetone (1:1) *Daucus carota* oil extract (DCOE)	Gavage, 20 μL of 100% oil Intraperitoneal, 0.3 mL of 2% oil. Topical, 0.2 mL of 100% oil; 50% oil; 5% oil.	Minimal effects seen with gavage administration. Significant decrease in tumor volume, delay in tumor appearance, and inhibition of tumor incidence and yield with intraperitoneal and topical administration.	[65]
		Chemoprevention: Pre-treatment with DCOE (25 mg/kg) a week prior to cancer induction Treatment twice weekly for 14 weeks. Chemotherapeutic: DCOE (25 mg/kg body weight; IP; thrice a week for 8 weeks).	Reduced tumor incidence. Protection against DMBA-induced toxicity. Significant inhibition of tumor volume. No decrease in body weight as compared to cisplatin.	[80]
	Pentane: diethyl ether (50:50) fraction (F2)	Intraperitoneal treatment (10–200 mg/kg)	Significant inhibition of papilloma incidence, yield, and volume at weeks 15, 18, and 21.	[91]
	β-2-himachalen-6-ol (HC)	Topical (5%). Intraperitoneal HC (25 mg/kg).	Significant decrease in papilloma yield and volume at weeks 12, 16, and 18, and increase in survival rates. No decrease in weight with HC (safer).	[93]
		Topical (5%). Intraperitoneal HC (10, 25, and 50 mg/kg).	Significant decrease in papilloma yield, incidence, and volume, and twofold to threefold increase in survival rates.	[95]

**Table 6 plants-13-00093-t006:** In vitro antibacterial effects of *D. carota* subspecies.

DC Subspecies	Plant Organ, Location	Treatment	Detail	Concentration/MICs	References
*carota*	Herbs and umbels, Poland	Essential oil (hydrodistillation)	Inhibitory effects against *Bacillus subtilis* and *Staphylococcus aureus*.	MIC = 3–5 μL/mL	[8]
			Inhibitory effects against *Escherichia coli* and *Pseudomonas aeruginosa*.	MIC = ≥8 μL/mL	
	Ripe and unripe fruits, flowers, roots, leaves. and stems; Serbia		All oils; strongest inhibitory effects against *Bacillus cereus*:	MIC = 5.0–50.0 μL/mL	[69]
			Ripe fruit oil;	MIC = 5.0–25.0 μL/mL	
			Unripe fruit oil;	MIC = 10.0–25.0 μL/mL	
			Flower oil;	MIC = 50.0–400.0 μL/mL	
			Root oil;	MIC = 5.0–150.0 μL/mL	
			Stem oil;	MIC = 25.0–250.0 μL/mL	
			Leaf oil.	MIC = 25.0–350.0 μL/mL	
	Ripe umbels, Portugal		*Bacillus subtilis*, *Listeria monocytogenes*, and *Staphylococcus aureus*.	MIC = 0.32–0.64 μL/mL	[66]
			*Escherichia coli* and *Salmonella typhimurium*.	MIC > 10 μL/mL	
	Leaves, flower, petals, and fruits; Uzbekistan		*Bacillus subtilis* and *Staphylococcus aureus*.	MIC = 6–7 μL/mL	[71]
	The aerial parts, France	Essential oil (vapor distillation)	Inhibitory effects against *C. jejuni*, *C. coli*, and *C. lari*.	MIC = 250 μg/mL	[5]
			Inhibitory effects against *C. jejuni* F38O11, LV7, and LV9 clinical isolates vs. human isolate *LV11*.	MIC = 250 µg/mL vs. MIC = 500 µg/mL	
			Inhibitory effects against *C. jejuni LM7A*.	MIC = 250 µg/mL	
			Inhibitory effects against *C. jejuni Lme27A*.	MIC = 125 µg/mL	
	Umbels, Tunisia	Essential oil (hydrodistillation) and supercritical CO_2_ extracts	Inhibitory effects against *Escherichia coli* ATCC 35218 and *Staphylococcus aureus* ATCC 43300.	MIC > 2.5% (*v*/*v*)	[9]
	Umbels, Lebanon	*Daucus carota* (DC) aqueous and methanolic extracts	Inhibitory effects against *Staphylococcus aureus* meti S and meti R:		[97]
			Aqueous extract;	MIC = 20 mg/mL	
			Methanolic extract.	MIC = 10 mg/mL	
*maritimus*	Flowers and roots, Tunisia	Essential oil (hydrodistillation)	Inhibitory effects against Gram-positive and Gram-negative bacteria.	MIC = 1.25–5 mg/mL	[76]
			Flower oil: more effective against *E. coli* (*ESLβ*).	MIC = 1.25 mg/mL	
			Root oil: more effective against *S. aureus*, *S. pneumonia*, *Shigella* spp., and *E. faecalis*.	MIC = 1.25 mg/mL	
*hispanicus*	Roots and aerial parts, Algeria		Aerial part oil:		[77]
			Inhibitory effects against *B. subtilis*;	MIC = 1.2 mg/mL	
			Inhibitory effects against *S. aureus*. Root oil:	MIC = 4.8 mg/mL	
			Inhibitory effects against *B. subtilis*;	MIC = 1.5 mg/mL	
			Inhibitory effects against *S. aureus*.	MIC = 4.2 mg/mL	
*maximus*	Green seeds, Italy	Essential oil (steam distillation)	Inhibitory effects against Gram-positive strains (*Staphylococcus* and six out of twenty-five *L. monocytogenes* strains) and Grams-negative strains (*Acinetobacter* and *Stenotrophomonas maltophilia* ICE272).	MIC = 1.25–2.5 μL/mL	[75]

**Table 7 plants-13-00093-t007:** In vitro antifungal effects of *D. carota* subspecies.

DC Subspecies	Plant Organ, Location	Treatment	Detail	Concentration/MICs	References
*carota*	Ripe and unripe fruits, flowers, roots, leaves, and stems; Serbia	Essential oil (hydrodistillation)	Inhibitory effects against *Fulvia fulvum*;	MIC = 2.0–100.0 μL/mL	[69]
			Inhibitory effects against *Trichoderma viride*;	MIC = 25.0–100.0 μL/mL	
			Inhibitory effects against *Aspergillus ochraceus*.	MIC = 10.0–150.0 μL/mL	
	Herbs and umbels, Poland		Inhibitory effects against *Candida albicans*;	MIC = 5 μL/mL	[8]
			Inhibitory effects against *Penicillium expansum*.	MIC = 8 μL/mL	
	Umbels, Tunisia		Inhibitory effects against clinical strains of *Candida albicans* and *C. tropicalis* 1011 RM.	MIC > 2.5% (*v*/*v*)	[9]
	Ripe umbels, Portugal		Inhibitory effects against *Cryptococcus neoformans*;	MIC = 0.16 μL/mL	[66]
			Inhibitory effects against dermatophytes;	MIC = 0.32–0.64 μL/mL	
			Inhibitory effects against *C. guilliermondii*.	MIC = 0.32 μL/mL	
	Fruits, Uzbekistan		Inhibitory effects against *Candida albicans*.	MIC = 12 μL/mL	[71]
	Blooming and flowering umbels, Sardinia Islands and Portugal	Essential oil (hydrodistillation) and supercritical CO_2_ extracts	Portugal:		[6]
			Inhibitory effects against *Cryptococcus neoformans*;	MIC = 0.32–0.64 μL/mL	
			Inhibitory effects against dermatophytes.	MIC = 1.25–2.5 μL/mL	
			Italy:		
			Inhibitory effects against *Cryptococcus neoformans*;	MIC = 0.32 μL/mL	
			Inhibitory effects against dermatophytes.	MIC = 0.16–0.32 μL/mL	
*gummifer*	Aerial parts, Portugal	Essential oil (hydrodistillation)	Inhibitory effects against dermatophyte strains and *C. neoformans*;	MIC = 0.32–0.64 μL/mL	[24]
			Inhibitory effects against *C. guillermondii*.	MIC = 1.25 μL/mL	
*halophilus*	Umbels, Portugal		Inhibitory effects against dermatophytes.	MIC = 0.16–0.64 μL/mL	[73]
*hispanicus*	Root and aerial parts, Algeria		Inhibitory effects against *Candida albicans*		[77]
			Aerial parts:	MIC = 0.078 ± 0.02 mg/mL	
			Roots:	MIC = 0.125 ± 0.04 mg/mL	
			Inhibitory effects against *Aspergillus flavus*.		
			Root vs. aerial parts oil inhibition:	100% vs. 42.22%	
*maximus*	Aerial parts, Portugal		Inhibitory effects against dermatophytes and *Cryptococus neoformans*;	MIC = 0.16–0.32 μL/mL	[74]
			Inhibitory effects against Aspergillus;	MIC = 0.32–0.64 μL/mL	
			Inhibitory effects against Candida.	MIC = 0.64–1.25 μL/mL	

## Data Availability

Not applicable.

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
