# Peer review of "The Wild Carrot (Daucus carota): A Phytochemical and Pharmacological Review"

_plants, 2023, doi:10.3390/plants13010093_

Round 1

Reviewer 1 Report

Comments and Suggestions for Authors

The paper “Wild carrot (Daucus carota): A phytochemical and pharmacological review” provides a comprehensive and detailed review of the phytochemical composition, traditional medicinal uses, and pharmacological properties of the wild carrot subspecies, Daucus carota. The authors have meticulously compiled and analyzed information from various electronic databases, covering literature from as far back as 1927 to early 2022, which underscores the depth and breadth of their research.

The highlight of this review is the extensive cataloging of over 310 compounds found in thirteen wild Daucus carota subspecies. Notably, the paper details the presence of terpenoids, phenylpropenoids, flavonoids, and phenolic acids, with a significant mention of 40 compounds constituting more than 3% of the overall composition. This level of detail is impressive and provides a rich resource for researchers interested in the phytochemical aspects of wild carrots.

Another significant contribution of this paper is the correlation of these phytochemicals with various pharmacological properties. The review adeptly discusses the antioxidant, anticancer, antipyretic, analgesic, antibacterial, antifungal, hypolipidemic, hepato-, and gastroprotective properties of these subspecies. This aspect of the study not only reinforces the traditional uses of wild carrots in treating infections, inflammation, and cancer but also opens avenues for further research in these areas. 

Nonetheless, there are several aspects of the paper that could benefit from further enhancement. Key considerations for future revisions include: 

In the manuscript, the reference to traditional knowledge in lines 62-64 could be made more precise by specifying which ancestral groups or cultures recognized the health benefits of both edible and wild carrot subspecies. The current phrasing, "our ancestors," is too broad and lacks cultural or geographical context. It would be more informative to identify specific ethnic groups or societies, thus contextualizing this ethnopharmacological knowledge more accurately.

There is a notable redundancy in lines 74 and 86. This repetition could be addressed by either merging the information in these lines for conciseness or by rephrasing and differentiating the content to maintain clarity and avoid duplication.

The methodology section requires further elaboration for a clearer understanding of the research's scope and depth. It should detail not only the keywords used and the time period covered but also the number of papers reviewed and their geographical distribution. This additional information would provide a more comprehensive understanding of the research methodology and the breadth of literature surveyed.

Regarding the ethnopharmacological use of Daucus carota, the review lacks clarity on the specific regions or cultures being examined. A clearer delineation of the geographical or cultural scope of the review, whether it pertains to the practices of Ancient Greeks, Romans, Lebanese, or other groups, is necessary for a more precise understanding. Including a table summarizing this information could greatly enhance the clarity and usability of the data.

Finally, the generalization of "traditional medicine" in the discussion section seems to lack specificity and depth. It is advisable to diversify the references used in this section to avoid repetition from chapter 4.

Both the discussion and conclusion chapters would benefit significantly from further development, including a more nuanced exploration of the traditional medicinal uses of Daucus carota in various cultures and a more detailed synthesis of the findings, their implications, and future research directions.

To conclude, the manuscript "Wild carrot (Daucus carota): A phytochemical and pharmacological review" exhibits potential for acceptance, contingent upon major revisions.

Comments on the Quality of English Language

Minor editing of English language required. 

Reviewer 2 Report

Comments and Suggestions for Authors

Abstract:

-line 16: Daucus carota L.

-line 17: instead (ssp. sativus), rewrite - (D. carota L. ssp. sativus (Hoffm.) Arcang.)

-line 22 instead: wild carrot subspecies (Daucus carota) rewrite - wild carrot subspecies (D. carota L. ssp. carota)

-lines 22 and 23: Various electronic 22 databases were consulted, and the literature from 1927 to early 2022 was reviewed?! It is necessary to update the information! Include the whole of 2022 and 2023 (2023 is almost finished as well, and it is necessary to have "fresh" data in order for the review paper to be relevant). The same comment is for Methodology (line 86).

General: Take care of the nomenclature! When the species is mentioned for the first time in the text, it is necessary to state the full name: for example Daucus carota L. (later in the text it is mentioned only as D. carota), Daucus carota L. ssp. sativus (Hoffm.) Arcang. (later in the text it is mentioned only as D. carota ssp. sativus), Daucus carota L. ssp. carota (D. carota ssp. carota), Daucus carota ssp. boissieri (Schweinf.) Hosni (D. carota ssp. boissieri), etc.

Round 2

Reviewer 1 Report

Comments and Suggestions for Authors

In the manuscript "Wild Carrot (Daucus carota): A Phytochemical and Pharmacological Review," the authors have successfully addressed several concerns previously raised. However, there remains a vital area needing enhancement:

The section detailing the methodology requires further clarification. It is currently not clear what specific keywords were employed during the literature search. Moreover, there is a need for explicit identification of the sources that were examined. Understanding whether a comprehensive review of scholarly papers was conducted is essential, along with the specific themes or topics that were focused on. Additionally, the criteria for selecting these particular works for analysis should be clearly outlined.

Providing these minor yet crucial revisions, especially in the methodology section, would significantly enhance the manuscript's clarity and rigor. Once these adjustments are made, the manuscript stands a good chance of being suitable for publication.

Comments on the Quality of English Language

Minor editing of the English language is still required. 

Reviewer 2 Report

Comments and Suggestions for Authors

Line 39: Daucus carota ssp. sativus (Hoffm.) Arcang. correct to D. carota ssp. sativus (Hoffm.) Arcang.

Line 40: Daucus carota ssp. boissieri correct to D. carota ssp. boissieri (Schweinf.) H. A. Hosni

Line 43: Daucus carota ssp. carota correct to D. carota ssp. carota

Line 44: Daucus carota correct to D. carota

Line 44-45: Daucus carota ssp. sativus correct to D. carota ssp. sativus

Line 46-47: Daucus carota L. coorect to D. carota

Line 55: Daucus carota correct to D. carota

Line 67: Daucus carota correct to D. carota

Line 74: Daucus carota correct to D. carota

Line 79: wild carrot (Daucus carota) - wild carrot (D. carota ssp. carota)

Same changes make through 3. Plant Description and Distribution, 4. Ethnopharmacological Use of Daucus carota, 5. Phytochemistry and Bioactive Compounds/Chemical Composition, 6. Pharmacological activities, 7. Safety/Toxicological Evaluation, 8. Discussion and Conclusion.
